# DAGE: DAG Query Answering via Relational Combinator with Logical Constraints

**Yunjie He** *University of Stuttgart, Bosch Center for Artificial Intelligence*
YUNJIE.HE@KI.UNI-STUTTGART
**Bo Xiong** *Stanford University*
**Daniel Hernández** *University of Stuttgart*
**Yuqicheng Zhu** *University of Stuttgart, Bosch Center for Artificial Intelligence*
**Evgeny Kharlamov** *Bosch Center for Artificial Intelligence*
**Steffen Staab** *University of Southampton, University of Stuttgart*

**Editors:** Leilani H. Gilpin, Eleonora Giunchiglia, Pascal Hitzler, and Emile van Krieken

## 1. Introduction to Problem

Predicting answers to queries over knowledge graphs is called a complex reasoning task because answering a query requires subdividing it into subqueries. To tackle this, query embedding (QE) methods (Ren et al., 2020; Zhang et al., 2021; Ren and Leskovec, 2020; Kotnis et al., 2021) encode queries as low-dimensional vectors and use neural logical operators to combine subquery embeddings into a single query embedding. However, these methods are limited to processing first-order logic queries with a single unquantified target variable, known as *tree-form queries*, which align with $\mathcal{SROI}^-$ description logic (He et al., 2024) and have tree-like computation graphs (Ren et al., 2023). In contrast, our work (He et al., 2025) explores *DAG queries*, a more expressive set that extends tree-form queries by enabling multiple paths from a quantified variable $x$ to a target variable $y$, unlike tree-form queries, which permit at most one such path.

Via an example, we illustrate the difference between DAG and tree-form queries and explain why existing query embedding methods cannot handle DAG queries. Consider the following first-order query $\phi(y)$, asking for works edited by an Oscar winner and produced by an Oscar winner.

$$\phi(y) ::= \exists x_1 \exists x_2 : \mathsf{wonBy}(\mathsf{Oscar}, x_1) \wedge \mathsf{edited}(x_1, y) \wedge \tag{1}$$
$$\mathsf{wonBy}(\mathsf{Oscar}, x_2) \wedge \mathsf{produced}(x_2, y).$$

Query $\phi(y)$ is tree-form because there exists at most one path from $x_1$ to $y$ and at most one path from $x_2$ to $y$. In $\mathcal{SROI}^-$, query $\phi(y)$ can be expressed as a concept description $C = C_1 \sqcap C_2$, where the decomposed subqueries $C_1 ::= \exists \mathsf{edited}^-.(\exists \mathsf{wonBy}^-.\{\mathsf{Oscar}\})$ and $C_2 ::= \exists \mathsf{produced}^-.(\exists \mathsf{wonBy}^-.\{\mathsf{Oscar}\})$. Existing query answering methods compute the embedding of query $\phi$ using the embeddings of two subqueries. They employ a neural or fuzzy logical operator to represent $\sqcap$, making query decomposition and logical connective modeling essential for existing reasoning approaches. However, this decomposition of queries does no longer hold for some DAG queries (tree-form is a special subcategory of DAG query). Consider the query asking for works edited and produced by an Oscar winner (i.e., an Oscar

winner that has both roles, editor and producer). Compared with the previous query, this new query enforces $x_1 = x_2$, which can be encoded by renaming both variables $x_1$ and $x_2$ as $x$:

$$\psi(y) = \exists x : \mathsf{wonBy}(\mathsf{Oscar}, x) \ \wedge \mathsf{edited}(x, y) \ \wedge \mathsf{produced}(x, y). \tag{2}$$

Query $\psi(y)$ cannot be decomposed into two tree-form queries because the conjunction between the query atoms $\mathsf{edited}(x, y)$ and $\mathsf{produced}(x, y)$ requires considering two target variables in the complex reasoning subtask. Thus, the existing methods cannot be applied to DAG queries. Instead, we can apply them by relaxing DAG queries to tree-form queries. However, one can expect that this workaround solution produces less accurate results because the solutions to the relaxed query $\phi(y)$ in (1) are not enforced to satisfy $x_1 = x_2$ like the solutions to query $\psi(y)$ in (2).

## 2. DAGE

In this section, we introduce DAGE (He et al., 2025), a novel extension of query embedding methods for DAG queries. The DAG query $\psi(y)$ is not expressible in $\mathcal{SROI}^-$ because $\mathcal{SROI}^-$ allows for conjunctions in concept descriptions but not role descriptions, which are required to indicate that $x$ "$\mathsf{produced}$ and $\mathsf{edited}$" $y$. A description logic that allows for conjunctions in role descriptions, called $\mathcal{ALCOIR}$, can express the query $\psi(y)$ as the following concept description $D$

$$D ::= \exists (\mathsf{edited} \sqcap \mathsf{produced})^-.(\exists \mathsf{wonBy}^-.\{\mathsf{Oscar}\}). \tag{3}$$

Unlike existing methods, to compute the embedding of query $D$, we do not decompose $D$ into two subqueries, but we propose to compute the embedding of the relation description $\mathsf{edited} \sqcap$ $\mathsf{produced}$ with an additional neural operator, called *relational combinator*, to represent the intersection between relations. To define the relational combinator for role conjunctions, we incorporate regularization terms based on $\mathcal{ALCOIR}$ tautologies into the learning objective to ensure model compliance.

## 3. Datasets and Primary Results

Existing complex query answering benchmark datasets lack DAG queries. To evaluate DAGE, we introduce six new DAG query types (2s, 3s, sp, us, is, ins) and generate three new benchmark datasets for the DAG query answering task: NELL-DAG, FB15k-237-DAG, and FB15k-DAG. Using existing query embedding methods, Query2Box (Ren et al., 2020), ConE (Zhang et al., 2021), and BetaE (Ren and Leskovec, 2020) as backbones, we implement DAGE for DAG query processing. Our experiment results and extensive analyses show that:

1. DAGE consistently delivers significant improvements to all baseline methods across all query types and datasets.

2. DAGE can handle DAG queries while still performing well on tree-form queries.

3. DAGE significantly improves baseline models, particularly on the challenging tasks they previously couldn't handle on their own.

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
