# OpenReview forum: "DAGE: DAG Query Answering via Relational Combinator with Logical Constraints"
_nesyconf.org/NeSy/2025/Conference_Phase_2 — NeSy 2025 - Phase 2 Poster_

### Official Review · Reviewer_M6pR · 2025-07-02
**Handling DAG-based queries in Query Embedding**

**Rating:** 7
**Confidence:** 3

**Review:**

The article summarizes the work on answering DAG-structured queries over knowledge graphs, a more expressive class of logical queries than standard tree-form queries handled by existing Query Embedding (QE) models.
The authors introduce DAGE, an extension that supports conjunctive roles by modeling intersections of relations via arelational combinator and enforcing logical consistency through regularization based on the ALCOIR description logic.
DAGE is evaluated on newly proposed datasets designed to benchmark DAG-style queries. Experimental results show consistent performance gains over baselines, especially in cases where previous models failed. DAGE also maintains strong performance on tree-form queries, demonstrating robustness and generality.

Minor comments:
- Could the numbers for the improvement over the baselines be provided? Does DAGE also improve performance over non-DAG queries? If so, how?

**Anonymity:**

Remain anonymous

---

### Official Review · Reviewer_7QL8 · 2025-07-07
**DAGE: DAG Query Answering via Relational Combinator with Logical Constraints, valid as extended abstract**

**Rating:** 7
**Confidence:** 4

**Review:**

This paper is an extended abstract. Below are details on what was required for this type of pater.

- Was the original paper published in a top-tier venue? Yes . The work summarised here is the paper “DAGE: DAG Query Answering via Relational Combinator with Logical Constraints,” published in the WWW ’25.

-  The paper extends neural query-embedding methods to handle general DAG queries by introducing a relational combinator regularised by description-logic tautologies. It explicitly bridges symbolic description-logic reasoning with neural embeddings. Therefore in topic with NeSy

- The abstract (i) motivates the gap (tree-form vs. DAG queries), (ii) outlines the method (relational combinator with logical constraints), and (iii) summarises empirical gains.

It is worth accepting it on NeSy as an extended abstract.

**Anonymity:**

Remain anonymous